# Dynamic control of gene regulatory logic by seemingly redundant transcription factors

**Zohreh AkhavanAghdam[†], Joydeb Sinha[†], Omar P Tabbaa[†], Nan Hao\***

Section of Molecular Biology, Division of Biological Sciences, University of California, San Diego, La Jolla, United States

**Abstract** Many transcription factors co-express with their homologs to regulate identical target genes, however the advantages of such redundancies remain elusive. Using single-cell imaging and microfluidics, we study the yeast general stress response transcription factor Msn2 and its seemingly redundant homolog Msn4. We find that gene regulation by these two factors is analogous to logic gate systems. Target genes with fast activation kinetics can be fully induced by either factor, behaving as an 'OR' gate. In contrast, target genes with slow activation kinetics behave as an 'AND' gate, requiring distinct contributions from both factors, upon transient stimulation. Furthermore, such genes become an 'OR' gate when the input duration is prolonged, suggesting that the logic gate scheme is not static but rather dependent on the input dynamics. Therefore, Msn2 and Msn4 enable a time-based mode of combinatorial gene regulation that might be applicable to homologous transcription factors in other organisms.

**\*For correspondence:** nhao@ucsd.edu

[†]These authors contributed equally to this work

**Competing interests:** The authors declare that no competing interests exist.

## Introduction

Homologous transcription factors (TFs) often co-exist in eukaryotic cells, resulting in seemingly redundant regulation of their target genes. Although a large number of TF homologs have diversified over time to obtain distinct target genes from their partners, others have remained relatively conserved and share the same DNA binding motif, which limits their downstream interactions to identical target genes. Recent studies suggest that some closely related TF homologs or isoforms, which regulate a shared set of target genes, might have diverged expression patterns, dynamic responses or gene regulatory functions. For example, the yeast transcriptional regulator Dig1 inhibits the expression of mating response genes to pheromone stimulation, whereas its homolog Dig2 exhibits both negative and positive regulation depending on the conditions (*Chou et al., 2008*; *Houser et al., 2012*). In mammalian cells, two TF isoforms NFAT1 and NFAT4 display distinct nuclear translocation dynamics in response to stimuli. It has been suggested that this dynamic diversity of isoforms might enhance the temporal signal processing function of the cell (*Yissachar et al., 2013*). In addition, a very recent study showed that the TF homologs STAT5A and STAT5B differentially contribute to the immune transcriptional response due to their different expression levels (*Villarino et al., 2016*). Here we use the yeast homologous stress responsive TFs Msn2 and Msn4 as a model to quantitatively study the functional relevance of closely related TFs in the same single cells.

Msn2 and Msn4 are $C_2H_2$ zinc-finger TFs that regulate cellular responses to a wide range of environmental stresses (*Schmitt and McEntee, 1996*). Upon stress stimulation, both TFs rapidly translocate from the cytoplasm to the nucleus where they bind to the same DNA recognition sequence and induce the expression of a common set of stress responsive genes (*Martinez-Pastor et al., 1996*). Their nucleocytoplasmic translocation is controlled by phosphorylation and is directly regulated by

**eLife digest** Cells respond to environmental signals by activating proteins called transcription factors. These bind to the DNA that is stored in the cell nucleus and turn on specific genes to make gene products. Many of these transcription factors move in and out of the nucleus once activated. Different environmental signals affect the amount of transcription factor that appears in the nucleus in different ways, and this is important in determining which genes should be turned on and how many copies of gene products should be made.

Many transcription factors co-exist with a similar version of themselves in the same cell. These closely related proteins, called homologous transcription factors, respond to the same signals and bind to the same place on the DNA to turn on the same genes. It was not clear what advantages the cells gain from having two molecules that perform the same roles.

Two homologous transcription factors called Msn2 and Msn4 are found in baker's yeast. These transcription factors respond to a wide variety of environmental stresses by moving rapidly into the nucleus, where they remain for a short time to turn on hundreds of target genes that are needed for the cell to survive.

AkhavanAghdam, Sinha, Tabbaa et al. investigated the roles of Msn2 and Msn4 by tracking where the proteins localized to and which genes they switched on inside the same single cell. Genes that can be turned on quickly could be activated by either Msn2 or Msn4, and both factors activated the genes to a similar extent. By contrast, both Msn2 and Msn4 were required to activate those genes that take a long time to be turned on. In these cases, Msn2 served as a 'switch' that governed the 'on' and 'off' state of the genes, while Msn4 behaved as a 'rheostat' to tune how much gene product was made. This cooperation between the two transcription factors is equivalent to a design commonly found in electrical circuits and may help the cell to survive in rapidly changing environments.

Further studies are now needed to investigate the mechanisms that provide Msn2 and Msn4 with distinct roles in gene regulation. Technological advances that allow the full genetic material of a single cell to be analyzed could also determine whether other homologous transcription factors regulate their target genes in similar ways.

protein kinase A (PKA) and phosphatases (*Gorner et al., 1998*) (*Figure 1A*, left). Therefore, it has been long believed that Msn2 and Msn4 are functionally redundant in regulating gene expression response. In fact, since Msn2 is assumed to play a more pronounced role in gene regulation, many previous studies focused only on Msn2, deleting the *MSN4* gene to simplify analysis (*Hansen and O'Shea, 2013*, *2015b*, *2016*; *Hao and O'Shea, 2012*; *Lin et al., 2015*; *Petrenko et al., 2013*). A microarray analysis, however, suggested that Msn2 and Msn4 might have different contributions to gene induction at individual promoters (*Berry and Gasch, 2008*), but the mechanism underlying these differences remains unknown.

Here, we combine quantitative single-cell imaging and high-throughput microfluidics to monitor and compare the dynamic responses and gene regulatory functions of Msn2 and Msn4 in single cells. We find that Msn2 and Msn4 have non-redundant and distinct functions in the combinatorial gene regulation. We have previously demonstrated that Msn2/4 target genes differ significantly in their promoter activation kinetics, which dramatically influences their responses to dynamic inputs (such as transient versus sustained inputs) (*Hao and O'Shea, 2012*). In this work, we show that, in response to a transient input, either Msn2 or Msn4 alone is sufficient to induce the expression of target genes with fast kinetics promoters, constituting what is essentially a biological 'OR' logic gate. In contrast, the induction of target genes with slow kinetics promoters requires activation of both factors, forming an 'AND' gate. At the single-cell level, even though Msn2 and Msn4 show similar nuclear translocation dynamics, they exhibit different levels of heterogeneity in nuclear localization and distinct gene regulatory functions. Msn2 is activated in a relatively homogeneous manner and functions as a low threshold 'switch' essential for turning on slow kinetics promoters. In contrast, Msn4 activation is highly heterogeneous and it serves as a 'rheostat' to effectively tune the induction level of target genes with slow kinetics promoters in individual cells. Therefore, while target genes

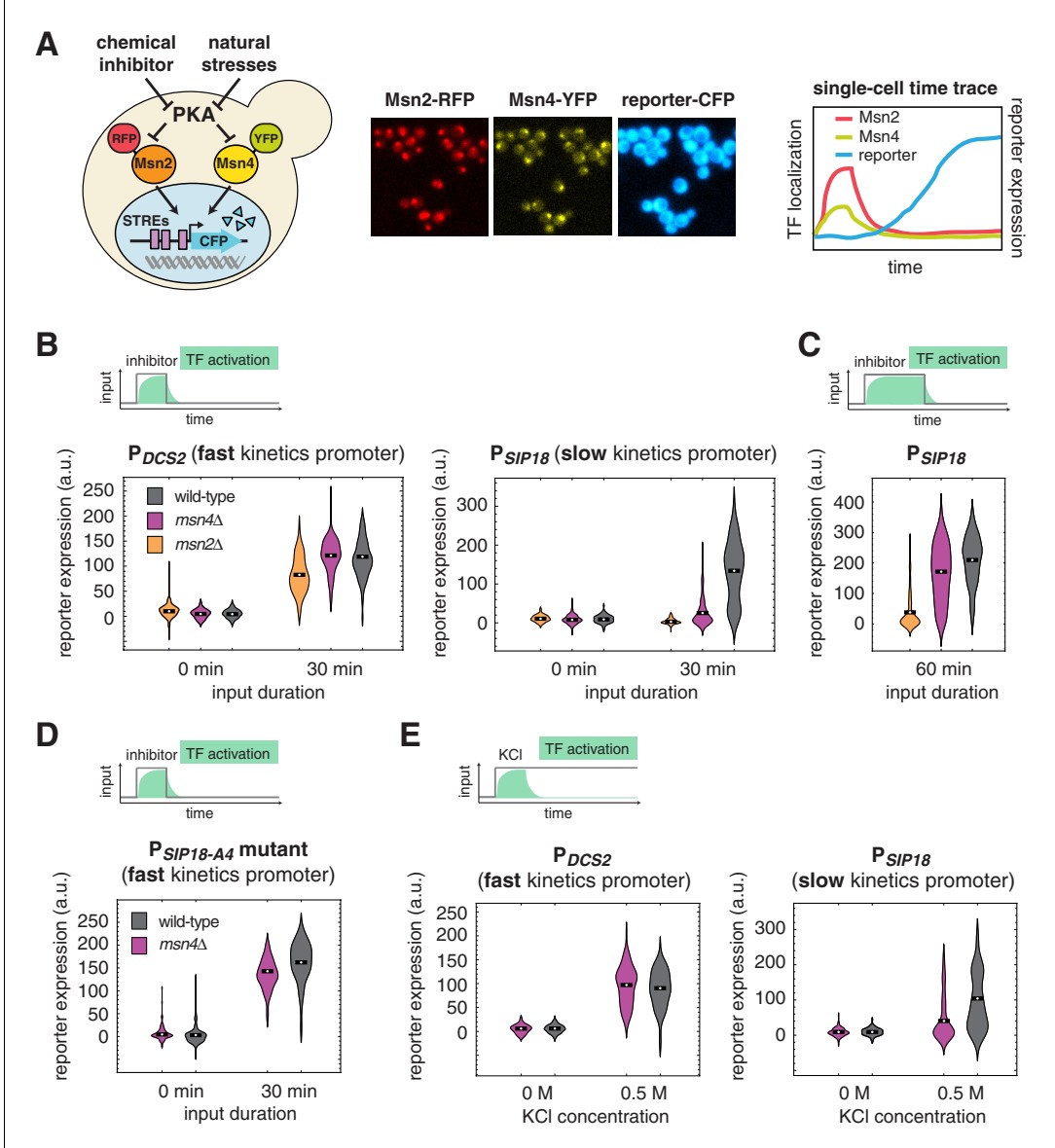

**Figure 1.** Msn4 is required for the induction of target genes with slow promoter kinetics. (**A**) Homologous TFs Msn2 and Msn4 are regulated by the same upstream PKA signals in response to natural stresses or chemical inhibitors and control a common set of target genes with stress response elements (STREs) in their promoters. In the same strain, Msn2 and Msn4 are fused with RFP and YFP respectively, at their native loci; a CFP reporter under the Msn2/4 specific promoter is introduced to monitor gene expression responses. Middle: Translocation of Msn2-RFP and Msn4-YFP and reporter gene expression can be monitored in the same single cells over time. Right: In response to stimulation, time traces of Msn2 and Msn4 translocation and reporter gene expression can be quantified for each single cell. For each condition, single-cell data are collected from at least three independent experiments. (**B**) Violin plots showing the distributions of reporter expression under (left) the fast kinetics promoter P$_{DCS2}$ or (right) the slow kinetics promoter P$_{SIP18}$ in single cells in response to 3 μM inhibitor inputs with 30-min pulse duration (illustrated by the top inset) in wild-type, msn2Δ, and msn4Δ strains, respectively (n: ~300 cells per condition per strain). The mean value of single cell responses was labeled using the black bar for each condition. The expression of the reporter gene was tracked in single cells over a 3-hr period in which the reporter fluorescence in most cells has already reached the plateau. The last point of each single-cell time trace was used in the plots (a.u.: arbitrary units). (**C**) Violin plots showing the distributions of reporter expression under the slow kinetics promoter P$_{SIP18}$ in response to a 60 min pulse of inhibitor input. (**D**) Violin plots showing the distributions of reporter expression under the faster mutant promoter P$_{SIP18-A4}$ in wild-type and msn4Δ strains, respectively, in response to 30-min inhibitor input. (**E**) Violin plots showing the distributions of reporter expression under (left) the fast kinetics promoter P$_{DCS2}$ or (right) the slow kinetics promoter P$_{SIP18}$ in response to 0.5 M KCl in wild-type and msn4Δ strains, respectively. The sustained KCl stimulation leads to a transient pulse of TF activation, as illustrated in the top cartoon panel.

The following figure supplements are available for figure 1:

*Figure 1 continued on next page*

*Figure 1 continued*

**Figure supplement 1.** Dynamic profiles of reporter gene expression.
**Figure supplement 2.** The dependence on Msn4 might expand generally to slow kinetics promoters.

with fast kinetics promoters are uniformly expressed in most cells, those with slow promoters are more likely to be fully induced in only a fraction of cells with high Msn4 activity. Our work reveals that the seemingly redundant TF Msn4 has a distinct gene regulatory role from its homolog Msn2 and enables diversified gene expression responses within a cell population, which might be beneficial for survival under rapidly changing environments.

## Results

### Msn4 is required for the induction of target genes with slow promoter kinetics

To investigate gene regulation by Msn2 and Msn4 in single cells, we fused Msn4 with a yellow fluorescent protein (YFP) and Msn2 with a red fluorescent protein (RFP) at their native loci. A cyan fluorescent protein (CFP) reporter under Msn2/4 specific target promoters was introduced into the same strain to monitor downstream gene expression. To understand gene responses to dynamic TF activation, we have previously developed a chemical genetics method for controlling the Msn2/4 nuclear localization using a small molecule 1-NM-PP1 that specifically inhibits protein kinase A (PKA) activity (*Hao et al., 2013*; *Hao and O'Shea, 2012*). Here, we combine this method with quantitative time-lapse microscopy and microfluidics (*Hansen et al., 2015*; *Hao et al., 2013*; *Hao and O'Shea, 2012*) to simultaneously track Msn2 and Msn4 localization and target gene expression in a large number of individual cells in response to dynamic inputs (*Figure 1A*). In each experiment, we measure single-cell responses over a 3-hr period, which is sufficient for the fluorescent gene expression reporter to reach the plateau in most cells (*Figure 1A*, right).

Our previous studies revealed that Msn2/4 target promoters can be characterized as having fast or slow activation kinetics relative to one another based on the time needed for their activation (*Hansen and O'Shea, 2013*; *Hao and O'Shea, 2012*). While target genes with fast kinetics promoters respond strongly to transient TF inputs, slow kinetics promoters, due to their long activation delay, filter out inputs with short durations. The activation kinetics of target promoters depends on their promoter architectures, in particular, the organization of TF binding sites and nucleosomes (*Hansen and O'Shea, 2015a*; *Hao and O'Shea, 2012*). To analyze dynamic gene regulation by Msn2 and Msn4, we focus here on two well-characterized promoters – $P_{DCS2}$ and $P_{SIP18}$, which are Msn2/4 specific (not induced in a *msn2∆ msn4∆* strain) (*Hansen and O'Shea, 2013*), and have been routinely used to represent Msn2/4 target promoters with fast ($P_{DCS2}$) or slow ($P_{SIP18}$) activation kinetics, respectively (*Hansen and O'Shea, 2013*, *2015a*, *2015b*, *2016*). The *DCS2* promoter can be activated 5 times faster than the *SIP18* promoter for a given TF input (*Hansen and O'Shea, 2013*).

To first determine the dependence of target gene expression on Msn2 and Msn4, we measure the induction of Msn2/4 target promoters in response to TF inputs with various durations in wild-type cells and cells lacking *MSN2* or *MSN4* and plotted the distributions of single-cell expression responses (*Figure 1*; The dynamic profiles of reporter gene expression are shown in *Figure 1—figure supplement 1*). We find that, in response to a transient inhibitor input (30 min), activation of either Msn2 or Msn4 is sufficient to fully induce the fast kinetics promoter $P_{DCS2}$ (*Figure 1B*, left). In contrast, the induction of slow kinetics promoter $P_{SIP18}$ requires activation of both Msn2 and Msn4: the absence of either factor abolishes the expression of reporter gene (*Figure 1B*, right). Interestingly, in response to a prolonged input pulse (60 min), while Msn2 is still needed for the induction of the slow gene promoter, Msn4 is no longer required (*Figure 1C*). These results suggest that Msn4 functions to shift the activation time-scales of slow kinetics promoters and thereby enables the induction of such promoters by transient inputs.

To determine whether the requirement of Msn4 for gene induction is specific to slow promoter kinetics, we employed a mutant of the $P_{SIP18}$ promoter ($P_{SIP18}$-A4), which has been converted to a

fast kinetics promoter by moving the Msn2/4 binding sites more adjacent to the TATA box (*Hansen and O'Shea, 2015a*). In accordance with the fast kinetics promoter P$_{DCS2}$, Msn4 is not needed for the expression of reporter gene under the P$_{SIP18}$ mutant promoter (*Figure 1D*). To further examine whether other target gene promoters have similar dependence to Msn4, we analyzed the responses of fast kinetics promoter P$_{DDR2}$ and slow kinetics promoter P$_{TKL2}$. Similar to the fast kinetics promoter P$_{DCS2}$, the expression of the P$_{DDR2}$ reporter gene does not require Msn4. In contrast, Msn4 is needed for the full induction of P$_{TKL2}$ in response to a transient input (*Figure 1—figure supplement 2*). These results suggest that the dependence on Msn4 might be a general feature of slow kinetics promoters. Finally, to determine whether target promoters would respond similarly when cells are faced with natural stressors, we monitored the reporter expression of P$_{DCS2}$ and P$_{SIP18}$, respectively, in response to osmotic stress, which leads to a transient pulse of TF activation (*Hao et al., 2013*; *Hao and O'Shea, 2012*). Consistent with the inhibitor experiments, while Msn4 is not critical for the expression of fast kinetics promoter P$_{DCS2}$, it is required for the full induction of slow kinetics promoter P$_{SIP18}$ (*Figure 1E*).

Therefore, our work reveals that, contrary to what has been previously believed, Msn4 is not redundant to its homolog Msn2 in regulating gene expression. In particular, while activation of either Msn2 or Msn4 is sufficient to trigger the expression of target genes with fast promoter kinetics, target genes with slow promoter kinetics depend on both Msn2 and Msn4 for their full induction in response to biologically relevant transient inputs.

## Msn4 displays heterogeneous nuclear translocation in single cells

Having established that Msn4 is not redundant to its homolog Msn2, we next investigate the dynamic and functional differences between the two factors at the single cell level that can account for their specific contributions to gene regulation. We first focus on the dynamics of Msn2 and Msn4 nuclear translocation. We observe that Msn2 and Msn4 show similar temporal dynamics of translocation in the same single cells in response to a transient inhibitor input (*Figure 2A*). However, we find that the level of Msn4 nuclear translocation is highly heterogeneous across single cells: some cells show high level of nuclear translocation, while other cells have very low localization levels. In contrast, the translocation levels of Msn2 are relatively homogeneous amongst individual cells. To illustrate the noise levels of Msn2 and Msn4 translocation, we plotted the standard deviation of their single-cell time traces scaled by the mean and reported the coefficient of variation (CV: standard deviation scaled by the mean) for the peak point of time traces. As shown in *Figure 2A(ii)*, Msn4 nuclear translocation exhibits a higher level of cell-cell variability than that of Msn2. To investigate the dynamics of Msn2 and Msn4 translocation in response to natural stresses, we subject yeast cells to osmotic stress and ethanol stress treatments. As shown in *Figure 2B and C*, osmotic stress elicits a transient pulse of Msn2 and Msn4 translocation, while ethanol stress induces sustained nuclear localization of Msn2 and Msn4. In response to either stress, Msn4 exhibits similar temporal dynamics of nuclear translocation to Msn2 in single cells, consistent with the inhibitor treatments. In addition, the level of Msn4 nuclear localization shows a higher degree of cell-cell heterogeneity than that of Msn2 under natural stress conditions, in accordance with the inhibitor treatments.

To determine the relative nuclear concentrations of Msn2-RFP and Msn4-YFP molecules at the single cell level, we performed a control experiment to obtain a scaling factor that normalizes the nuclear fluorescence intensities of YFP and RFP into 'normalized a.u.' (*Figure 2—figure supplement 1A*). We observe that the nuclear localization level of Msn4 is generally lower (~3-fold lower) than that of Msn2 in the same cells under inhibitor or natural stress conditions (*Figure 2—figure supplement 1B*, yellow curves versus red curves). In addition, although Msn4 has a higher coefficient of variation, the standard deviation of Msn4 nuclear localization in single cells (without being scaled by the mean) is lower than that of Msn2 (*Figure 2—figure supplement 2*). These results suggest that the high degree of cell–cell variability of Msn4 might be largely due to its relatively low nuclear levels compared to that of Msn2.

Thus, our single-cell imaging analysis shows that, in response to various stimuli, nuclear translocation of Msn4 temporally correlates with that of Msn2 in the same cells; however, the level of Msn4 nuclear localization in individual cells is more heterogeneous than that of its homolog.

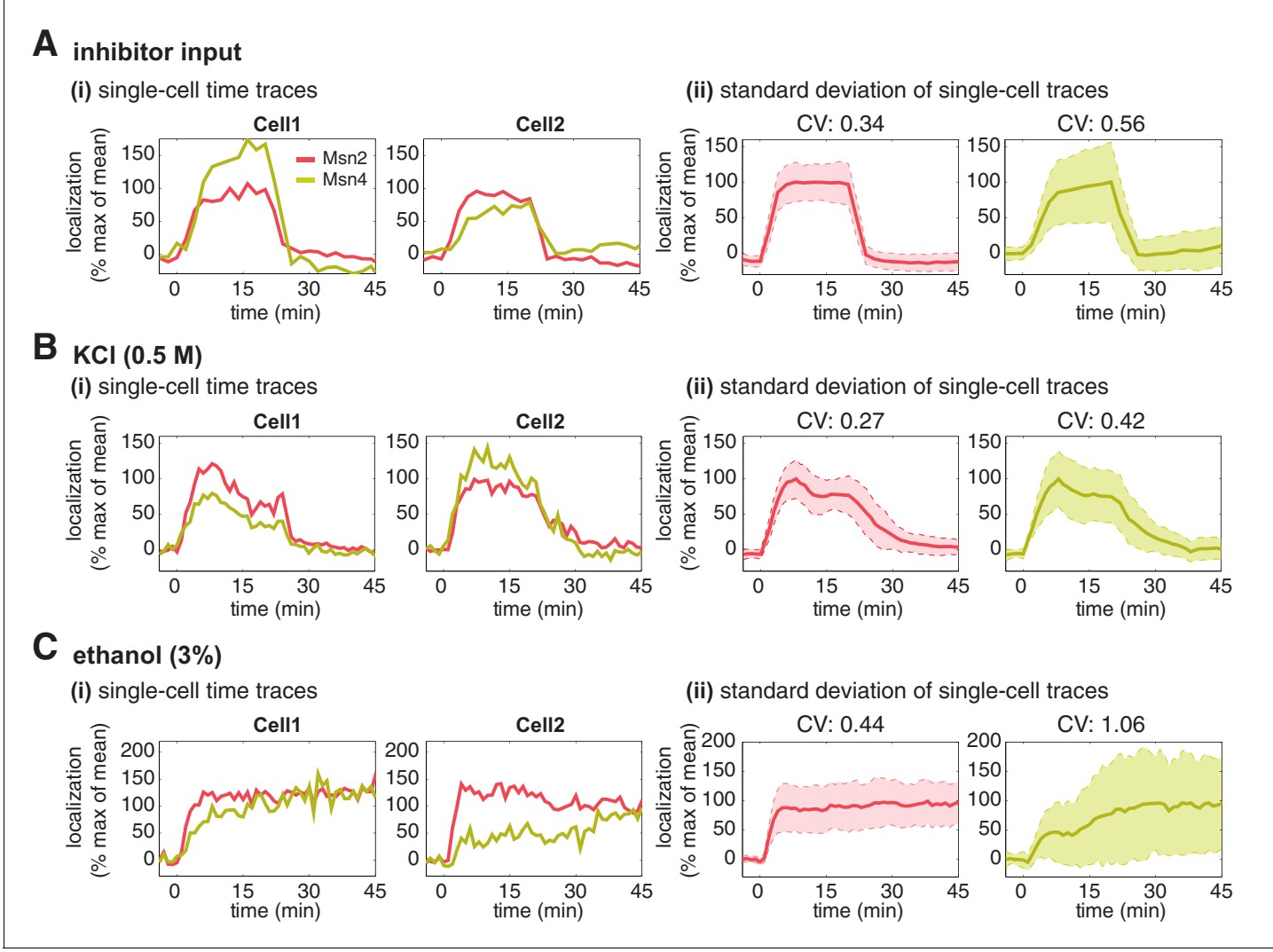

**Figure 2.** Msn2 and Msn4 show different levels of heterogeneity in single cells. Time traces of Msn2 and Msn4 nuclear translocation in the same single cells in response to (**A**) 20-min 1 μM inhibitor pulse, (**B**) 0.5M KCl, or (**C**) 3% ethanol. In each panel, (i) representative single-cell time traces of Msn2 and Msn4 nuclear translocation in the same single cells; (ii) standard deviation of single-cell time traces. For each condition, the single-cell time traces and standard deviations of single cell responses are scaled by the peak value of the averaged time traces (% max of mean). In (ii), the solid curve represents the averaged time trace; the shaded region represents the scaled standard deviation of single cell responses. The coefficient of variation (CV; the standard deviation divided by the mean) is calculated for the peak time point of time traces for each condition and displayed above each time trace.

The following figure supplements are available for figure 2:

**Figure supplement 1.** A direct comparison of the levels of Msn2 and Msn4 nuclear localization in the same cells.

**Figure supplement 2.** Single-cell time traces of Msn2 and Msn4 after normalization of YFP and RFP fluorescence.

## Msn4 exhibits distinct gene regulatory functions from Msn2 in single cells

Given that Msn2 and Msn4 show different levels of cell-cell variability in nuclear translocation, we speculate that they may play different regulatory roles in controlling heterogeneous gene expression at the single cell level. Using the deletion strains, we have revealed that target genes with different promoter activation kinetics exhibit different dependence to Msn2 and Msn4 (**Figure 1**). To further determine the dependence of gene regulation specifically on Msn2 or Msn4 in single cells when

both factors are present, we simultaneously monitored nuclear localization of Msn2 and Msn4 and reporter gene expression under the fast kinetics promoter $P_{DCS2}$ or the slow kinetics promoter $P_{SIP18}$ in the same wild-type cells. We then quantified and plotted the maximal level of reporter expression versus the peak nuclear localization level of Msn2 and/or Msn4 of each single cell to analyze the relationship between gene expression and the activity of Msn2 and Msn4, respectively. To cover a full range of TF translocation levels, we combined the single cell responses to 30-min inhibitor inputs with various doses.

We find that, for the fast kinetics promoter $P_{DCS2}$, gene expression can be induced in the majority of cells in which either Msn2 or Msn4 is adequately activated (*Figure 3A,i*). The level of reporter expression shows a similar graded relationship with both Msn2 and Msn4, reaching the saturation when either factor is activated over a low threshold level (*Figure 3Aii*; Single-cell distributions of gene expression versus Msn2 or Msn4 are shown in *Figure 3—figure supplement 1A*; The probabilities of gene induction versus Msn2 or Msn4 are shown in *Figure 3—figure supplement 2A*). These results are consistent with our observation by deletion analysis that Msn2 and Msn4 play largely redundant roles in regulating target genes with fast kinetics promoters (*Figure 1*).

In contrast, for the slow kinetics promoter $P_{SIP18}$, gene expression is highly heterogeneous among single cells, consistent with previous results (*Hansen and O'Shea, 2013*). Furthermore, the reporter gene is fully induced predominantly in the fraction of cells in which Msn4 is highly activated (*Figure 3B,i*, red solid circles). In individual cells with a fixed level of Msn2 activity, a higher level of Msn4 activation results in an increase in both the probability and the level of gene induction (*Figure 3B,i*, with a fixed y-axis value and an increasing x-axis value). However, in single cells with a fixed level of Msn4 activity, higher Msn2 activation does not necessarily lead to higher gene induction; in fact, too much Msn2 activation will suppress gene induction in the same cell, suggesting a competing role of Msn2 for binding to the promoter (*Figure 3B,i*, with a fixed x-axis value and an increasing y-axis value). To quantitatively demonstrate this competing role of Msn2 against Msn4, we plotted the relationship between gene expression and the ratio of Msn2 versus Msn4 in single cells. As shown in *Figure 3—figure supplement 3*, gene expression decreases dramatically when the ratio of Msn2 versus Msn4 increases. We further analyzed the relationship between reporter gene expression with Msn2 or Msn4 activity, individually. Gene expression shows a switch-like relationship to Msn2 activation with a low threshold (~10 normalized a.u.: ~25% of maximal Msn2 localization): while a low level of Msn2 activity is required for turning the gene on, the induction level is independent of Msn2 activity (*Figure 3B,ii*, left, averaged response of single cells binned based on their TF levels). In contrast, gene expression exhibits a linear relationship with Msn4 activity in which both the probability and the level of gene induction increase with the level of Msn4 activity in single cells (*Figure 3B,ii*, right; Single-cell distributions of gene expression versus Msn2 or Msn4 are shown in *Figure 3—figure supplement 1B*; The probabilities of gene induction versus Msn2 or Msn4 are shown in *Figure 3—figure supplement 2B*). In accordance with the deletion analysis in *Figure 1C*, in response to prolonged (60-min) input pulses, the reporter expression of $P_{SIP18}$ no longer specifically depends on Msn4 and the level of reporter expression shows graded relationships with both Msn2 and Msn4 (*Figure 3—figure supplement 4*).

These results demonstrate that Msn2 and Msn4 play distinct and cooperative regulatory roles in controlling target genes with slow promoter kinetics. In response to transient inputs, consistent with the required role of Msn2 for slow promoter induction shown in *Figure 1*, Msn2 in single cells serves as a low threshold 'switch' for gene induction: it is required to be activated above a certain threshold (~25% of its maximal level) to turn on transcription in the cell; once its activity is above that threshold, a further increase in Msn2 activity cannot positively contribute to the extent of gene induction. In contrast, despite of its low expression level, Msn4 is a highly potent activator of slow target promoters and thus is also required the full induction of slow promoters (*Figure 1*). Once Msn2 turns on gene transcription in a cell, Msn4 functions as a 'rheostat' in the same cell to effectively fine-tune the probability and level of gene induction. Furthermore, high levels of Msn2 in the nucleus may compete with Msn4 for binding to the same target promoters and can thus suppress gene induction. This possible inhibitory role of Msn2 against Msn4 counteracts its modest positive contribution to gene expression. Therefore, the expression level of target genes with slow promoter kinetics depends specifically on the Msn4 activity in individual cells. Furthermore, the homolog-specific gene regulation depends on the transient dynamics of TF inputs. In response to sustained inputs, the slow

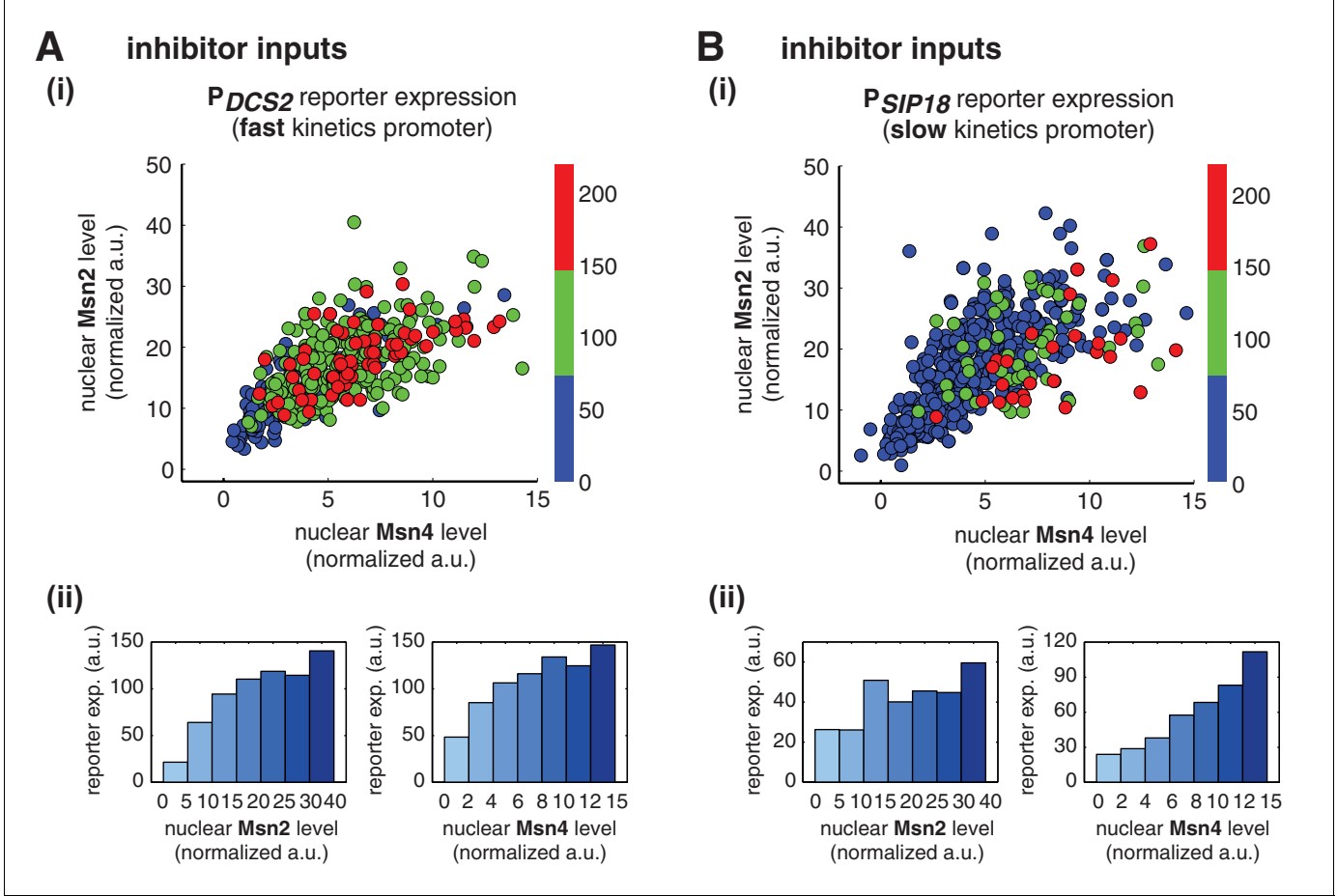

**Figure 3.** Msn2 and Msn4 exhibit distinct gene regulatory functions in single cells in response to 30-min inhibitor inputs. (A) (i) A scatter plot showing the relationship of the fast kinetics promoter P$_{DCS2}$ reporter expression with Msn2 and Msn4 activation at the single cell level. Each dot represents a single cell. Single-cell time traces were tracked over a 3-hr period in which the reporter fluorescence in most cells has already reached the plateau. The x and y axes represent the peak values of Msn4 and Msn2 nuclear translocation (the maximal values in the first 30 min of translocation time traces), respectively; and the dot color represents the maximal level of gene expression as indicated in the color bar. To cover the full dynamic range of TF translocation, the data from the experiments using 30 min inhibitor pulses with 0.1, 0.25, 0.5, 0.75 and 1 µM doses have been combined (n: 444 cells). (ii) Plots show the relationships between P$_{DCS2}$ reporter expression and (left) Msn2 or (right) Msn4, respectively. Single cells are binned based on their Msn2 or Msn4 nuclear level as indicated in the x-axis and the average of reporter expression is calculated for each binned groups of single cells and shown in the bar graphs. (B) Scatter plots and bar graphs showing the relationship of the slow kinetics promoter P$_{SIP18}$ reporter expression with Msn2 and Msn4 activation at the single cell level. The data analysis and presentation schemes are consistent with those in (A) (n: 595 cells). Single-cell data used in these plots are provided in the source data files.

The following source data and figure supplements are available for figure 3:

**Source data 1.** Source data for Figure 3A.
**Source data 2.** Source data for Figure 3B.
**Figure supplement 1.** Single-cell distributions of reporter gene expression versus nuclear TF levels in response to 30-min inhibitor inputs.
**Figure supplement 2.** The relationship between the probability of reporter gene expression and nuclear TF levels in response to 30-min inhibitor inputs.
**Figure supplement 3.** Relationship between P$_{SIP18}$ reporter gene expression and the ratio of nuclear Msn2 versus Msn4 in response to 30-min inhibitor inputs.
**Figure supplement 4.** Msn2 and Msn4 exhibit similar gene regulatory functions in single cells in response to 60-min inhibitor inputs.

*Figure 3 continued*

**Figure supplement 5.** Relationship between reporter gene expression and the area-under-the-curve (AUC) of nuclear TF levels in response to 30-min inhibitor inputs.

**Figure supplement 6.** Single-cell distributions of reporter gene expression versus the speed of TF nuclear import or export.

kinetics promoters no longer exhibit the specific dependence on Msn4 and behave like fast kinetics promoters upon transient inputs.

Finally, to evaluate the influence of other dynamic characteristics (such as nuclear import rate or export rate) of Msn2 or Msn4 translocation, we also plotted the level of gene expression versus the area under the curve (AUC) of Msn2 or Msn4 nuclear localization. As shown in *Figure 3—figure supplement 5*, the relationship between gene expression and AUC are similar to those shown in *Figure 3*. In accordance, because the time differences in nuclear import or export for Msn2 or Msn4 among single cells (~1–2 min) are relatively small comparing to the total duration of inputs (30 min), we observed modest influence on gene expression from the variations in these characteristics under our experimental conditions (*Figure 3—figure supplement 6*).

## Msn4 controls the induction level of slow kinetics promoters in single cells under natural stress conditions

To determine whether Msn4 functions differently from Msn2 under natural stress conditions, we treated yeast cells with osmotic stress (0.5 M KCl) or ethanol stress (4% ethanol), respectively, and measured nuclear localization of Msn2 and Msn4 and reporter gene expression under the slow kinetics promoter $P_{SIP18}$ in the same cells. In response to 0.5 M KCl treatment, we find that the reporter gene is fully induced specifically in the cells with high Msn4 activity (*Figure 4A,i and ii*), in accordance with the inhibitor experiments. In addition, the probability and the level of gene induction are independent of Msn2 activity, but increase linearly with Msn4 activity in single cells (Single-cell distributions of gene expression versus Msn2 or Msn4 are shown in *Figure 4—figure supplement 1A*; the relationship between gene expression and AUC are shown in *Figure 4—figure supplement 2A*). Because the osmotic stress treatment elicits Msn2 nuclear translocation above the threshold required for gene induction (~10 normalized a.u.: ~25% of maximal Msn2 localization; determined from the inhibitor experiments in *Figure 3*) in almost all of the cells, we could not observe the basal 'off' state of gene expression when we binned the cells with their Msn2 levels (*Figure 4A, ii*, left). This result indicates that, in response to the stress condition, Msn2 is 'switched on' in all of the cells and the induction level of slow kinetics promoters in individual cells is primarily controlled by Msn4 activity.

Similarly, in response to 4% ethanol treatment, the reporter gene is also induced specifically in cells with high Msn4 activity. The treatment of ethanol stress is a relatively harsh stress condition that leads to a global translational arrest (*Ding et al., 2009*; *Stanley et al., 2010*). As a result, the majority of cells are not able to express the reporter gene under the slow kinetics promoter; however, the cells that do express the reporter gene are those with high Msn4 activity (*Figure 4B,i*). Similar to the osmotic stress condition, Msn2 is activated above the threshold required for gene induction in most cells; therefore, the probability of gene expression does not show any dependence on Msn2 activity in single cells (*Figure 4B,ii*, left). In contrast, the probability of gene expression shows a linear relationship with Msn4 activity in single cells (*Figure 4B,ii*, right; Single-cell distributions of gene expression versus Msn2 or Msn4 are shown in *Figure 4—figure supplement 1B*; the relationship between gene expression and AUC are shown in *Figure 4—figure supplement 2B*), consistent with the inhibitor and osmotic stress experiments.

Taken together, these results show that, under natural stress conditions, Msn4 plays a distinct regulatory role from Msn2 in controlling target genes with slow kinetics promoters. Similar to inhibitor treatments, expression of these genes depends specifically on the Msn4 activity in a linear fashion at the single-cell level. Given its heterogeneous activity in single cells and its critical role in gene

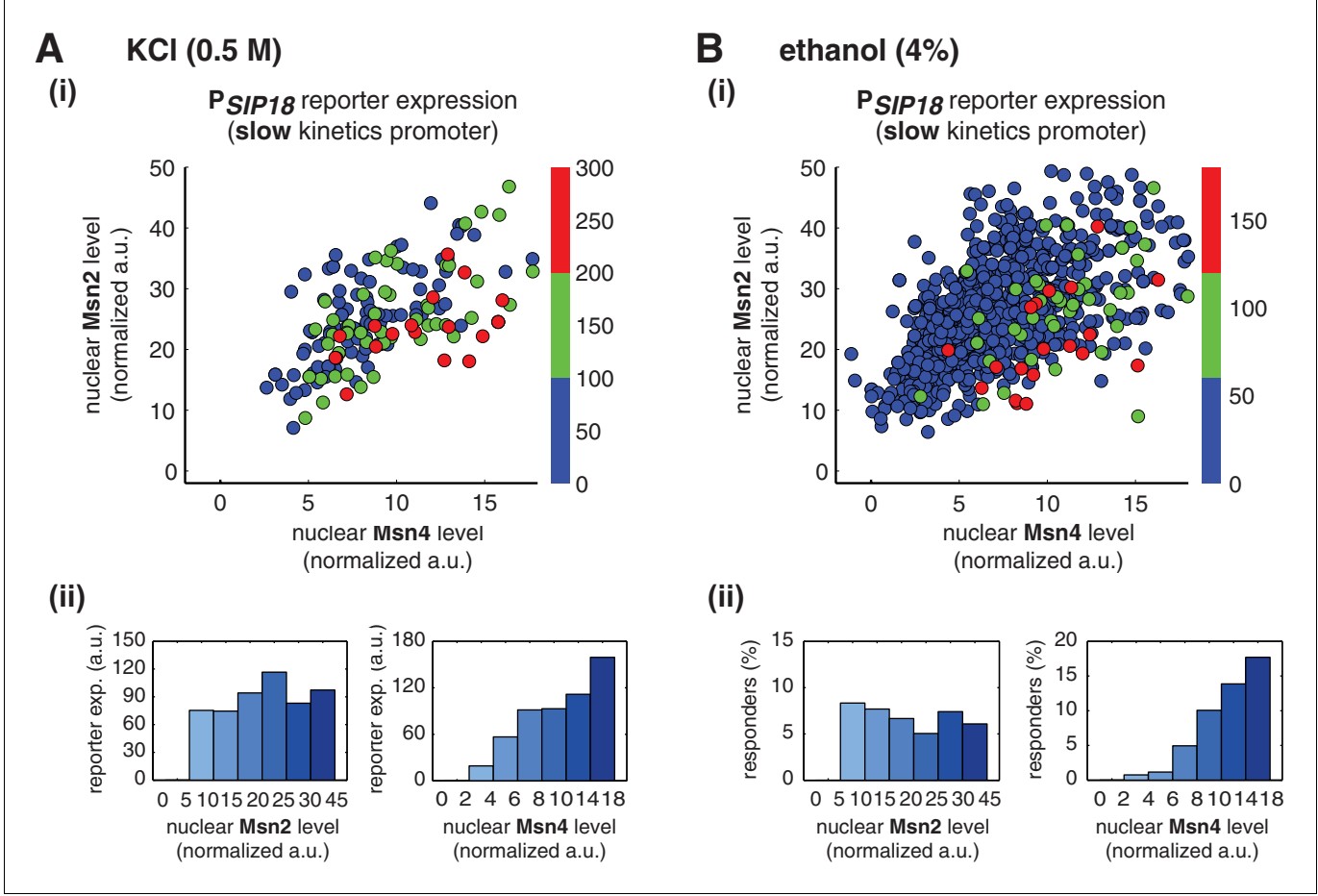

**Figure 4.** Msn2 and Msn4 exhibit distinct gene regulatory functions in single cells in response to natural stresses. (**A**) (i) A scatter plot showing the relationship of the slow kinetics promoter P$_{SIP18}$ reporter expression with Msn2 and Msn4 activation at the single cell level in response to 0.5M KCl. Each dot represents a single cell. The x and y axes represent the peak values of Msn4 and Msn2 nuclear translocation (the maximal values in the first 30 min of translocation time traces), respectively; and the dot color represents the peak level of gene expression as indicated in the color bar (n: 182 cells). (ii) Plots show the relationships between P$_{SIP18}$ reporter expression and (left) Msn2 or (right) Msn4, respectively. Single cells are binned based on their Msn2 or Msn4 nuclear level as indicated in the x-axis and the average of reporter expression is calculated for each binned groups of single cells and shown in the bar graphs. (**B**) Scatter plots and bar graphs showing the relationship of the slow kinetics promoter P$_{SIP18}$ reporter expression with Msn2 and Msn4 activation at the single cell level in response to 4% ethanol. The data analysis and presentation schemes are consistent with those in (**A**). Because the majority of cells are not able to express the reporter gene, the proportion of 'responder' cells (green and red cells) is quantified and shown in the bar graphs, instead of the average of reporter expression. Data from a large number of single cells are collected to obtain enough responders (n: 924 cells). Single-cell data used in these plots are provided in the source data files.

The following source data and figure supplements are available for figure 4:

**Source data 1.** Source data for Figure 4A.
**Source data 2.** Source data for Figure 4B.
**Figure supplement 1.** Single-cell distributions of reporter gene expression versus nuclear TF levels in response to natural stresses.
**Figure supplement 2.** Relationship between reporter gene expression and the area-under-the-curve (AUC) of nuclear TF levels in response to natural stresses.

regulation, Msn4, working in parallel with its homolog Msn2, can diversify the expression of a specific group of target genes within a cell population in response to natural stresses.

## Distinct gene regulatory functions of Msn2 and Msn4 might extend more generally to other target promoters

To determine whether Msn2 and Msn4 exhibit distinct regulatory functions on promoters other than $P_{DCS2}$ and $P_{SIP18}$, we measured the nuclear localization of Msn2 and Msn4 and reporter gene expression under fast kinetics promoter $P_{DDR2}$ and slow kinetics promoter $P_{TKL2}$. We find that, similar to the fast kinetics promoter $P_{DCS2}$, expression of the $P_{DDR2}$ reporter gene can be induced in most cells in which either Msn2 or Msn4 is activated over a low threshold level and the level of reporter expression shows a similar graded relationship with both Msn2 and Msn4 (*Figure 5A*). In contrast, the slow kinetics promoter $P_{TKL2}$ shows a similar response to that of $P_{SIP18}$, in which reporter expression is specifically induced in the fraction of cells with high Msn4 activity. The level of reporter expression shows a switch-like relationship to Msn2 activity, but a linear relationship with Msn4 activity (*Figure 5B*). These results suggest that the distinct functions of Msn2 and Msn4 in combinatorial gene regulation might be applicable to other target genes.

To further examine the generality of distinct regulatory functions of Msn2 and Msn4, we analyzed the gene expression response under the mutated $P_{SIP18}$ promoter ($P_{SIP18}$-A4), which, by incorporating a few mutations, has been converted from a slow kinetics promoter to a fast kinetics promoter (*Hansen and O'Shea, 2015a*). As shown in *Figure 5C*, in accordance with other fast kinetics promoters, gene expression under this mutant promoter is induced in most cells and displays a similar graded relationship with both Msn2 and Msn4. Therefore, Msn2 and Msn4 no longer show distinct regulatory functions when the $P_{SIP18}$ promoter, with the majority of the promoter sequence intact, is mutated to obtain fast activation kinetics. We next tested the opposite situation in which we slow down a fast kinetics promoter. To this end, we monitored the gene expression response under the $P_{DCS2}$ promoter in cells lacking the SWI/SNF chromatin remodeling complex (*snf6Δ*). It has been shown previously that this mutant significantly slows down the activation kinetics of fast promoters (*Hansen and O'Shea, 2013*). Consequently, we observe that gene expression under the $P_{DCS2}$ promoter in this mutant exhibits switch-like versus linear relationships with Msn2 and Msn4, respectively (*Figure 5D*), consistent with the responses of other slow kinetics promoters. In summary, these results suggest that the distinct regulatory functions of Msn2 and Msn4 might depend more generally on the kinetics of promoter activation, but not on specific target promoters.

In this study, we have focused on a few representative promoters because our single-cell analysis requires live-cell time-lapse experiments, the throughput of which hinders the examination of gene regulation at a more global level. However, we anticipate that, in near future, the technological advances will allow us to track single-cell gene regulation at the whole genome level. Such technologies will undoubtedly provide further insights into combinatorial gene regulation by Msn2 and Msn4. For example, we previously grouped target genes with fast versus slow promoter kinetics based on a population-level assay using cells with the *msn4Δ* background (*Hao and O'Shea, 2012*). Given the newly identified role of Msn4 in shifting the promoter activation timescales in a subpopulation of cells, we suspect that its presence might alter the classification of some target genes with intermediate promoter kinetics. A genome-wide single-cell analysis will enable a more accurate classification of target genes and, more importantly, will lead to a comprehensive understanding about dynamic regulation of global transcriptional responses to environmental stimuli.

## Discussion

Here we show that the homologous TFs Msn2 and Msn4, which have long been assumed to be functionally redundant, play distinct roles in coordinating differential expression of target genes depending on their promoter kinetics. For target genes with fast promoter kinetics, both factors can contribute to gene expression in a graded manner. In contrast, for target genes with slow promoter kinetics, Msn2 and Msn4 play distinct and cooperative roles, in which Msn2 functions as a low threshold 'switch' governing the 'ON' and 'OFF' state of promoter activation, while Msn4 serves as a 'rheostat' to effectively tune the induction level of gene expression (*Figure 6A*). Further biochemical analysis is needed to elucidate the mechanistic details underlying these distinct regulatory functions of Msn2 and Msn4 at slow target promoters. One possible mechanism could involve the recruitment

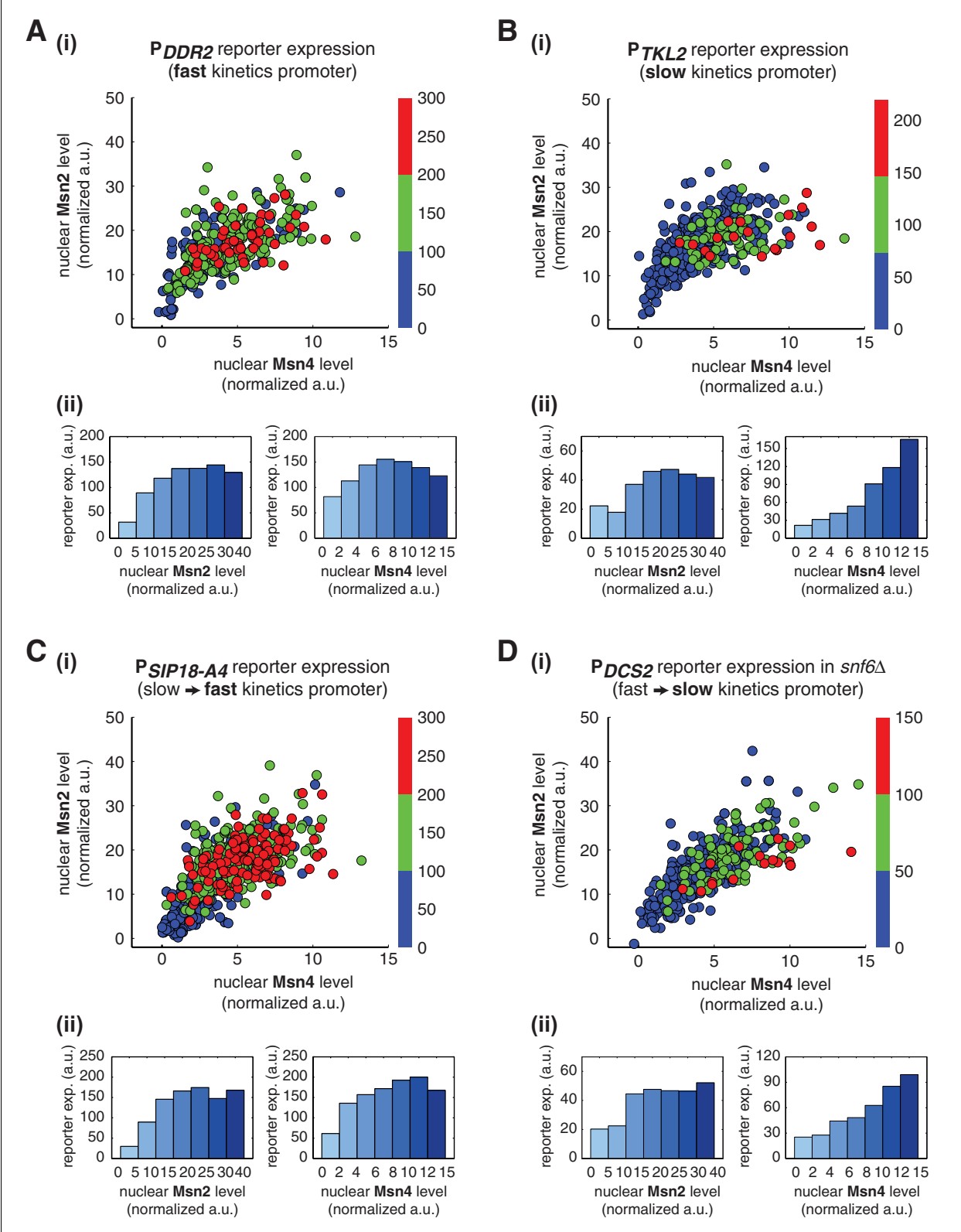

**Figure 5.** Gene regulatory functions of Msn2 and Msn4 on other fast or slow kinetics promoters. (**A**) (i) A scatter plot showing the relationship of the fast kinetics promoter P$_{DDR2}$ reporter expression with Msn2 and Msn4 activation at the single cell level. Each dot represents a single cell. Single-cell time traces were tracked over a 3-hr period in which the reporter fluorescence in most cells has already reached the plateau. The x and y axes represent the peak values of Msn4 and Msn2 nuclear translocation (the maximal values in the first 30 min of translocation time traces), respectively; and the dot color

*Figure 5 continued on next page*

*Figure 5 continued*

represents the maximal level of gene expression as indicated in the color bar. To cover the full dynamic range of TF translocation, the data from the experiments using 30 min inhibitor pulses with 0.1, 0.25, 0.5, 0.75 and 1 μM doses have been combined (n: 407 cells). (ii) Plots show the relationships between $P_{DDR2}$ reporter expression and (left) Msn2 or (right) Msn4, respectively. Single cells are binned based on their Msn2 or Msn4 nuclear level as indicated in the x-axis and the average of reporter expression is calculated for each binned groups of single cells and shown in the bar graphs. Scatter plots and bar graphs showing the relationship between gene expression and Msn2 and Msn4 activation for (**B**) the slow kinetics promoter $P_{TKL2}$ (n: 476 cells), (**C**) the promoter mutant $P_{SIP18-A4}$ (n: 553 cells), and (**D**) the promoter $P_{DCS2}$ in *snf6Δ* (n: 352 cells). The data analysis and presentation schemes are consistent with those in (**A**).

The following source data is available for figure 5:

**Source data 1.** Source data for Figure 5A.
**Source data 2.** Source data for Figure 5B.
**Source data 3.** Source data for Figure 5C.
**Source data 4.** Source data for Figure 5D.

of distinct chromatin remodeling factors by Msn2 and Msn4 to target gene promoters. For example, to function as a low threshold 'switch', Msn2 might first recruit some initiation factors critical for opening up the tightly packed nucleosomes, characteristic of slow kinetics promoters (*Hansen and O'Shea, 2013*, *2015a*; *Hao and O'Shea, 2012*). This could be followed by the subsequent promoter binding of Msn4, leading to the recruitment of Msn4-specific chromatin remodelers or modifiers to effectively promote and stabilize chromatin disassembly. In accordance with this mechanism, we observe that Msn4 always follows Msn2 in nuclear translocation with a short delay (~2–3 min) under the inhibitor or natural stress conditions (*Figure 2—figure supplement 1C* and *Figure 3—figure supplement 6*).

Target genes with fast and slow promoter activation kinetics are regulated differently and hence might have distinct physiological functions. In support of this, we find a close correlation between gene functions and promoter kinetics for previously identified target gene groups with fast or slow kinetics promoters (*Figure 6—figure supplement 1*; target gene groups are from *Hao and O'Shea, 2012*). Target genes with fast kinetics promoters are primarily involved in metabolic and cellular adaptation to glucose starvation (carbohydrate metabolism and autophagy), whereas the majority of target genes with slow kinetics promoters are involved in cellular protection against chronic stresses. These results suggest that, in response to carbon source changes that might require an immediate adaptive response, cells could quickly modulate their metabolism via activating those genes with fast kinetics promoters. In contrast, the induction of genes with slow kinetics promoters is more tightly controlled. These genes are important for preparing cells to survive under chronic stress conditions, the response to which might be less time-sensitive and the occurrence of which might be less frequent in natural habitats. Cells would only activate these genes upon a sustained presence of stresses. This temporal separation of target genes with different functions could avoid initiating resource-intensive cell protection processes in response to minor environmental fluctuations and thereby optimize resource allocation under rapidly changing environments. Furthermore, cells may use Msn4 to control the level of heterogeneity at the population level (*Figure 6B*), as part of a bet hedging strategy against unpredictable environmental conditions. During the first onset of stresses, a subpopulation of cells with high Msn4 activity induce the expression of stress resistance genes with slow kinetics promoters, preparing for upcoming severe or chronic stresses; meanwhile, other cells with low Msn4 activity cannot induce these genes and therefore may consequently obtain a better fitness advantage if the subsequent stress is minor or short term. In this way, cells can be divided into two subpopulations, each of which is specialized at coping with one of the possible environmental scenarios. Cells lacking Msn4, however, are not able to induce the slow target genes within the whole population, and hence might be more likely to go extinct in the face of extreme stresses. Therefore, the Msn4-dependent gene regulation may represent a strategy that enables a homolog-'controlled' form of heterogeneity within a cell population.

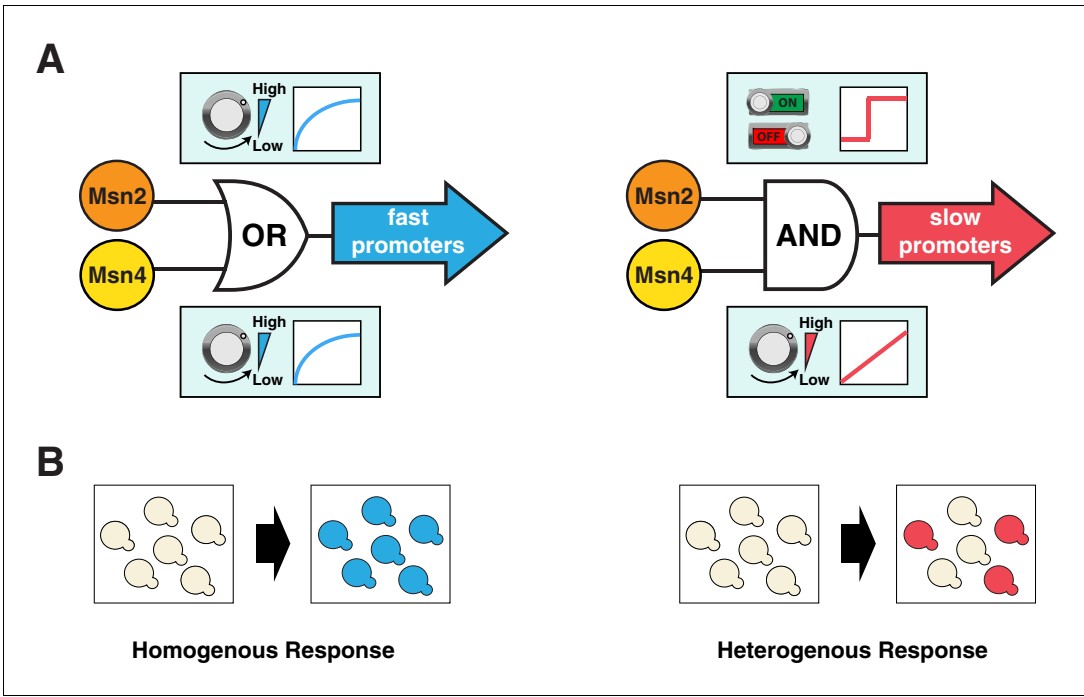

**Figure 6.** Schematics of the gene regulatory logic by Msn2 and Msn4. (**A**) Diagrams illustrating the gene regulatory schemes of Msn2 and Msn4 in controlling (left) fast or (right) slow kinetics promoters. Left: Either Msn2 or Msn4 is sufficient for the induction of fast promoters, constituting an 'OR' logic gate. At the single cell level, gene expression shows a similar graded dependence on both Msn2 and Msn4 and reaches saturation upon a low TF activity. Right: Msn2 and Msn4 are both required for the induction of slow promoters, constituting an 'AND' logic gate. At the single cell level, Msn2 serves as a low threshold 'switch' turning transcription ON or OFF depending on its activity. In contrast, Msn4 functions as a 'rheostat', tuning the gene induction level in a linear fashion. (**B**) Diagrams illustrating how the gene regulatory schemes of Msn2 and Msn4 contribute to the heterogeneity in gene expression at the population level. Left: an 'OR' logic gate will lead to homogeneous gene expression in a cell population. Right: an 'AND' logic gate with the 'rheostat' TF Msn4 produces a heterogeneous response in a population of cells.

The following figure supplement is available for figure 6:

**Figure supplement 1.** Biological functions of target genes with fast or slow kinetics promoters.

The coupling of the homologous factors Msn2 and Msn4 in combinatorial gene regulation is analogous to logic gate systems commonly found in digital circuits: fast kinetics promoters behave as an 'OR' gate, becoming fully induced with adequate amount of either factor, while slow kinetics promoters behave as an 'AND' gate, requiring the activation of both Msn2 and Msn4 (*Figure 6*). Interestingly, this logic gate scheme is not fixed, but rather dependent on the upstream dynamics of TF input – an 'AND' gate upon a transient input can become an 'OR' gate when the input duration is prolonged. As shown in *Figure 1B and C*, in response to a 30-min input pulse, the slow kinetics promoter P$_{SIP18}$ is an 'AND' gate, requiring both Msn2 and Msn4 for gene induction; But it becomes an 'OR' gate in response to a 60-min input pulse, in which Msn4 is no longer required. Therefore, given enough amount of time in the nucleus, Msn2 can also function as a 'rheostat' to compensate the absence of Msn4 and tune the induction level of slow target promoters, consistent with a previous study showing that increasing the steady-state expression level of Msn2 leads to a graded induction of its target genes (*Stewart-Ornstein et al., 2013*). Our results suggest that the architectures of gene regulatory networks are not static and could be rewired by various upstream dynamics of TF inputs. In yeast, a recent proteomic analysis found that most proteins that exhibit transient pulsatile dynamics to environmental changes are members of paralogous or closely related TFs (*Dalal et al., 2014*). For example, Msn2 and a related transcriptional repressor Mig1, both having pulsatile

dynamics, regulate their common target genes by modulating their relative pulse timing (*Lin et al., 2015*). Moreover, in mammalian systems, an increasing number of TFs, including some closely related TF pairs such as NFAT1 and NFAT4 (*Yissachar et al., 2013*), have been identified to possess highly diverse activation dynamics that contribute to gene expression responses (*Behar and Hoffmann, 2010*; *Purvis et al., 2012*; *Purvis and Lahav, 2013*; *Tay et al., 2010*; *Werner et al., 2005*). Given the prevalence of seemingly redundant TFs in eukaryotes, we anticipate that the time-dependent combinatorial gene regulation revealed here for Msn2 and Msn4 will be widely applicable to homologous or closely related TFs that are controlled dynamically in other organisms including mammals.

# Materials and methods

## Yeast strain construction

Standard methods for the growth, maintenance and transformation of yeast and bacteria and for manipulation of DNA were used throughout. All *Saccharomyces cerevisiae* strains used in this study are derived from the W303 background (*ADE+ MATa trp1 leu2 ura3 his3 can1 GAL+ psi+*). A list of strains is provided in *Table 1*.

Msn2 was C-terminally tagged with a yeast codon-optimized mCherry by replacing the endogenous stop codon of the *MSN2* locus with *URA3* and then replacing the *URA3* with a linker-mCherry PCR fragment from a pKT vector using 5-FOA. Msn4 was C-terminally tagged with a linker-mCitrineV163A PCR fragment generated from a pKT vector containing yeast codon-optimized mCitrine with the V163A mutation to allow for fast maturation. The endogenous *MSN2* and *MSN4* terminators were left unchanged. *MSN2* and *MSN4* deletion strains (*msn2Δ* and *msn4Δ*, respectively) were made by replacing the endogenous *MSN2* or *MSN4* ORF with *TRP1*. The introduction of gene expression

**Table 1.** Yeast strains used in this work.

| Strain Name | Description |
|---|---|
| NH0084 | *NHP6a-IRFP:kanMX, TPK1$^{M164G}$, TPK2$^{M147G}$, TPK3$^{M165G}$* |
| NH0094 | *MSN4-mCitrineV163A, MSN2-mCherry, NHP6a-IRFP:kanMX, TPK1$^{M164G}$, TPK2$^{M147G}$, TPK3$^{M165G}$* |
| NH0095 | *msn4Δ::TRP1, MSN2-mCherry, NHP6a-IRFP:kanMX, TPK1$^{M164G}$, TPK2$^{M147G}$, TPK3$^{M165G}$* |
| NH0108 | *MSN4-mCitrineV163A, msn2Δ::natMX, NHP6a-IRFP:kanMX, TPK1$^{M164G}$, TPK2$^{M147G}$, TPK3$^{M165G}$* |
| NH0096 | *msn4Δ::TRP1, msn2Δ::natMX, NHP6a-IRFP:kanMX, TPK1$^{M164G}$, TPK2$^{M147G}$, TPK3$^{M165G}$* |
| NH0116 | *P$_{SIP18}$-mTurqouise2-HIS, MSN4-mCitrineV163A, MSN2-mCherry, NHP6a-IRFP:kanMX, TPK1$^{M164G}$, TPK2$^{M147G}$, TPK3$^{M165G}$* |
| NH0117 | *P$_{SIP18}$- mTurqouise2-HIS, msn4Δ::TRP1, MSN2-mCherry, NHP6a-IRFP:kanMX, TPK1$^{M164G}$, TPK2$^{M147G}$, TPK3$^{M165G}$* |
| NH0119 | *P$_{SIP18}$- mTurqouise2-HIS, MSN4-mCitrineV163A, msn2Δ::natMX, NHP6a-IRFP:kanMX, TPK1$^{M164G}$, TPK2$^{M147G}$, TPK3$^{M165G}$* |
| NH0120 | *P$_{DCS2}$- mTurqouise2-HIS, MSN4-mCitrineV163A, MSN2-mCherry, NHP6a-IRFP:kanMX, TPK1$^{M164G}$, TPK2$^{M147G}$, TPK3$^{M165G}$* |
| NH0121 | *P$_{DCS2}$- mTurqouise2-HIS, msn4Δ::TRP1, MSN2-mCherry, NHP6a-IRFP:kanMX, TPK1$^{M164G}$, TPK2$^{M147G}$, TPK3$^{M165G}$* |
| NH0110 | *P$_{DCS2}$- mTurqouise2-HIS, MSN4-mCitrineV163A, msn2Δ::natMX,, NHP6a-IRFP:kanMX, TPK1$^{M164G}$, TPK2$^{M147G}$, TPK3$^{M165G}$* |
| NH0333 | *P$_{SIP18–A4}$- mTurqouise2-HIS, MSN4-mCitrineV163A, MSN2-mCherry, NHP6a-IRFP:kanMX, TPK1$^{M164G}$, TPK2$^{M147G}$, TPK3$^{M165G}$* |
| NH0334 | *P$_{SIP18–A4}$- mTurqouise2-HIS, msn4Δ::TRP1, MSN2-mCherry, NHP6a-IRFP:kanMX, TPK1$^{M164G}$, TPK2$^{M147G}$, TPK3$^{M165G}$* |
| NH0335 | *P$_{SIP18-A4}$- mTurqouise2-HIS, MSN4-mCitrineV163A, msn2Δ::natMX, NHP6a-IRFP:kanMX, TPK1$^{M164G}$, TPK2$^{M147G}$, TPK3$^{M165G}$* |
| NH0425 | *P$_{TKL2}$- mTurqouise2-HIS, MSN4-mCitrineV163A, MSN2-mCherry, NHP6a-IRFP:kanMX, TPK1$^{M164G}$, TPK2$^{M147G}$, TPK3$^{M165G}$* |
| NH0426 | *P$_{TKL2}$- mTurqouise2-HIS, msn4Δ::TRP1, MSN2-mCherry, NHP6a-IRFP:kanMX, TPK1$^{M164G}$, TPK2$^{M147G}$, TPK3$^{M165G}$* |
| NH0427 | *P$_{DDR2}$- mTurqouise2-HIS, MSN4-mCitrineV163A, MSN2-mCherry, NHP6a-IRFP:kanMX, TPK1$^{M164G}$, TPK2$^{M147G}$, TPK3$^{M165G}$* |
| NH0428 | *P$_{DDR2}$- mTurqouise2-HIS, msn4Δ::TRP1, MSN2-mCherry, NHP6a-IRFP:kanMX, TPK1$^{M164G}$, TPK2$^{M147G}$, TPK3$^{M165G}$* |
| NH0459 | *P$_{DCS2}$- mTurqouise2-HIS, MSN4-mCitrineV163A, MSN2-mCherry, NHP6a-IRFP:kanMX, TPK1$^{M164G}$, TPK2$^{M147G}$, TPK3$^{M165G}$, snf6::cgURA3* |
| NH0237 | *MSN2- mCitrineV163A -HIS, NHP6a-IRFP:kanMX* |
| NH0267 | *MSN2- mCherry –TRP1, NHP6a-IRFP:kanMX* |

reporters into yeast was performed as described previously (*Hansen and O'Shea, 2013*). The fluorescence reporter gene used is a yeast codon-optimized mTurqouise2.

## Microfluidics

The microfluidics device used in this study is modified from a previously reported device (*Hersen et al., 2008*). The mask was designed to allow for bonding of two antiparallel Y-shaped devices on a single microfluidics chip using standard methods of soft lithography and replica molding. The device fabrication and the setup of microfluidic experiments were performed as described previously (*Hansen et al., 2015*; *Hao et al., 2013*; *Hao and O'Shea, 2012*).

## Time-lapse microscopy

All time-lapse microscopy experiments were performed using a Nikon Ti-E inverted fluorescence microscope with Perfect Focus, coupled with an EMCCD camera (Andor iXon X3 DU897). The light source is a Spectra X LED system. Images were taken using a CFI Plan Apochromat Lambda DM 60X Oil Immersion Objective (NA 1.40 WD 0.13MM). During experiments, the microfluidic device was taped to a customized device holder inserted onto the motorized stage (with encoders) of the microscope. For the experiments only tracking TF dynamics, three positions were chosen for each channel and the microscope was programmed to acquire Phase, YFP, mCherry, and iRFP images every two minutes. For the experiments measuring reporter gene expression, six positions were chosen for each experiment and the microscope was programmed to take iRFP and YFP or mCherry images every two minutes and both Phase and CFP images every 14 min for a total of three hours. In all experiments, cells in the device were first exposed to SD media for at least 30 min. When the image acquisition started, cells were maintained in SD media for the first five minutes to obtain a baseline for each fluorescence channel prior to the introduction of any stressor or 1-NM-PP1. The exposure and intensity settings for each channel were set as follows: CFP 300 ms at 9% lamp intensity, YFP 400 ms at 20% lamp intensity, mCherry 300 ms at 10% lamp intensity, and iRFP 200 ms at 15% lamp intensity. The camera was set to an EM Gain of 300 (within the linear range) for all four fluorescence channels.

## Image analysis

Fluorescence microscopy image stacks were pre-processed using ImageJ for background subtraction. The images were then processed using a custom MATLAB code for single-cell tracking and fluorescence quantification as described previously (*Hao et al., 2013*; *Hao and O'Shea, 2012*). We determine the sample size of our single-cell data based on similar studies published previously (*Hansen and O'Shea, 2013*; *Hao et al., 2013*; *Hao and O'Shea, 2012*).

## Acknowledgements

We thank Anders S Hansen, Jeff Hasty, Roy Wollman and Gurol M Süel for insightful suggestions and critical comments on the manuscript. This work was supported by NIH R01 GM111458.

## Additional information

### Funding

| Funder | Grant reference number | Author |
|---|---|---|
| National Institutes of Health | R01 GM111458 | Nan Hao |

The funders had no role in study design, data collection and interpretation, or the decision to submit the work for publication.

### Author contributions

ZA, JS, OPT, Conception and design, Acquisition of data, Analysis and interpretation of data, Drafting or revising the article; NH, Conception and design, Analysis and interpretation of data, Drafting or revising the article

**Author ORCIDs**

Nan Hao, http://orcid.org/0000-0003-2857-4789

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
