## [Decision Letter]

Thank you for submitting your article "Dynamic Control of Gene Regulatory Logic by Seemingly Redundant Transcription Factors" for consideration by *eLife*. Your article has been reviewed by three peer reviewers, and the evaluation has been overseen by a Reviewing Editor and Aviv Regev as the Senior Editor. The reviewers have opted to remain anonymous.

The reviewers have discussed the reviews with one another and the Reviewing Editor has drafted this decision to help you prepare a revised submission.

Summary:

Paralogous transcription factors are ubiquitous in many gene regulatory systems. However, how paralogous TFs quantitatively and specifically influence gene expression remains largely unexplored. Using single-cell time-lapse microscopy, this manuscript analyzes the regulation of genes by a pair of dynamic paralogous TFs, Msn2 and Msn4, in budding yeast. The authors use artificial induction and natural stress signals to transiently induce the system. They then measure TF dynamics and target gene outputs in the same cell, and analyze the quantitative roles of Msn2 and Msn4 in target gene expression levels. The data indicate that under certain conditions Msn2 acts as a switch while Ms4 behaves like a rheostat, and in other conditions Msn2/Msn4 function as an OR gate. These results suggest that dynamic control of these two stress response regulators may provide important physiological benefits for budding yeast in response to environmental signals.

Essential revisions:

1) Influence of the choice of metrics of Msn2/4 on "switch" versus "rheostat" behavior: In Figure 3–Figure 4, the authors plotted the relationship between target expression level vs. Msn2/Msn4 levels. In doing so, they chose "peak values of Msn4 and Msn2 nuclear translocation" as the "nuclear Msn2/Msn4 levels". This is not the most obvious or reasonable choice, since there are many characteristics of the Msn2/4 pulse that can be influencing gene expression downstream. For example, since the authors used total reporter output (i.e., peak CFP level) as expression level, shouldn't the corresponding input variable be the integrated area under the Msn2 or Msn4 nuclear localization curve? Therefore, the authors should report analysis with area under curve as "nuclear Msn2/Msn4 levels" and make sure that the switch or rheostat behavior for Msn2 or Msn4 remains unchanged. Also, the speed of Msn2/Msn4 nuclear translocation or exit could be influential, and the authors should report relationship for this characteristic of the pulse.

2) Generality of different regulatory functions of Msn2/4: The authors argue that the regulatory functions of Msn2 and Msn4 are different in different promoters (fast vs. slow). And in Figure 3, they show such differences by using one fast promoter (DSC2) and one slow promoter (SIP18). Since this is an essential data for the paper, other promoters should be reported. It seems like authors already have data with one other slow promoter (TKL2) as well as one other fast promoter (SIP18-A4). If so, these data should also be reported. The authors should also consider reporting a more global view of gene expression to support this important claim. Related to this point is that fact that the choice of fast and slow genes is based on classification made in a previous work of the author (Hao and O'Shea 2011), where it was done on the background of msn4 deletion. Given the critical role of Msn4 found here, it is possible that this classification can change. The authors should address this point, either through data analysis/reanalysis or discussion of relevant caveats.

3) Static versus dynamic profiles of gene expression: The authors only report the final level of the target genes (at 30 or 60 minutes) but not the target genes dynamic profiles (as demonstrated in Figure 1, right, blue trace). With these data, the msn4 mutant has a non-responsive Sip18 after 30 minutes, yet responsive at 60 minutes. Previous data from the π (Hao & O'Shea 2012, Supplementary Figure 12) suggests Sip18 does respond and climb steadily in this strain, albeit more weakly/slowly. Is it possible then that the msn4 mutant kinetics of Sip18 (and other "slow" targets) is simply slower/weaker, rather than qualitatively different ("AND" gate becoming an "OR" gate at prolonged inputs, as suggested by the authors)? The authors should therefore report target gene single cell trace, and use it to provide a more holistic interpretation of the results shown in Figure 1, and Figure 3.

4) Lack of quantitative data for some claims: Given the quantitative nature of this paper, some claims that could be made quantitatively are reported qualitatively. Results: "suggesting a competing role of Msn2 for binding to the promoter". The authors should make a plot to show the difference quantitatively. Similarly Results: "probability and the level of gene induction increase […]". There is no data pointing to the "probability" (or at least it is close to impossible to figure out the probability from the distribution data). Results: "much more heterogeneous". The authors should put numbers to the claim and probe whether this due to the relative small copy number of Msn4 compared to Msn2. Lastly, Discussion: "2-3min". If one normalizes the data (min to max) in Figure 2—figure supplement 1, Msn2 and Msn4 traces may simply overlay on top of each other (i.e., no time difference).

5) Functional analysis of promoters: Based on the authors' analysis, it seems like slow and fast promoters are regulated differently and can have distinct physiological functions. To link these mechanistic findings to potential functional and physiological implications, the authors should provide analyses supporting such roles, for example, showing correlation between induction dynamics and regulation by Msn2/4 to functional classification. If no such relationship exists, authors should also indicate so. Related to this point, promoters with fast and slow activation kinetics have their own advantages during adaptation upon stress stimulation. The authors should include an introduction or conclusion discussing in some detail "how the two mechanisms mediated by the two kinds of promoters are important for adaptation…".

6) Connection to osmotic stress: The reduced resistance of primed msn4 mutant cells to osmotic stress is intriguing, but the connection to the main story is not entirely clear. Should the authors opt to include this data, they should also include msn2 cells also respond to this stress under the same experimental conditions/protocols in order to probe whether this effect is Msn4 specific.

---

## [Author Response]

Essential revisions:

*1) Influence of the choice of metrics of Msn2/4 on "switch" versus "rheostat" behavior: In Figure 3–Figure 4, the authors plotted the relationship between target expression level vs. Msn2/Msn4 levels. In doing so, they chose "peak values of Msn4 and Msn2 nuclear translocation" as the "nuclear Msn2/Msn4 levels". This is not the most obvious or reasonable choice, since there are many characteristics of the Msn2/4 pulse that can be influencing gene expression downstream. For example, since the authors used total reporter output (i.e., peak CFP level) as expression level, shouldn't the corresponding input variable be the integrated area under the Msn2 or Msn4 nuclear localization curve? Therefore, the authors should report analysis with area under curve as "nuclear Msn2/Msn4 levels" and make sure that the switch or rheostat behavior for Msn2 or Msn4 remains unchanged. Also, the speed of Msn2/Msn4 nuclear translocation or exit could be influential, and the authors should report relationship for this characteristic of the pulse.*

We now include new Figure 3—figure supplement 5 and Figure 4—figure supplement 2 to show the relationship between gene expression and the AUC of TF nuclear translocation. As shown in these figures, we observed similar switch or rheostat behavior for Msn2 or Msn4 using AUC to represent the nuclear localization. We also include new Figure 3—figure supplement 6 to show the relationship between gene expression and the speed of nuclear import and export. Because the time differences in nuclear import or export for Msn2 or Msn4 among single cells (calculated and shown in Figure 3—figure supplement 6) are relatively small comparing to the total duration of inputs, we observed modest influence on gene expression from the variations in these characteristics.

In the main text, we add a new paragraph describing the results:

“Finally, to evaluate the influence of other dynamic characteristics (such as nuclear import rate or export rate) of Msn2 or Msn4 translocation, we also plotted the level of gene expression versus the area under the curve (AUC) of Msn2 or Msn4 nuclear localization. As shown in Figure 3—figure supplement 5, the relationships between gene expression and AUC are similar to those shown in Figure 3. In accordance, because the time differences in nuclear import or export for Msn2 or Msn4 among single cells (~1 – 2 minutes) are relatively small comparing to the total duration of inputs (30 minutes), we observed modest influence on gene expression from the variations in these characteristics under our experimental conditions (Figure 3—figure supplement 6).”

*2) Generality of different regulatory functions of Msn2/4: The authors argue that the regulatory functions of Msn2 and Msn4 are different in different promoters (fast vs. slow). And in Figure 3, they show such differences by using one fast promoter (DSC2) and one slow promoter (SIP18). Since this is an essential data for the paper, other promoters should be reported. It seems like authors already have data with one other slow promoter (TKL2) as well as one other fast promoter (SIP18-A4). If so, these data should also be reported. The authors should also consider reporting a more global view of gene expression to support this important claim. Related to this point is that fact that the choice of fast and slow genes is based on classification made in a previous work of the author (Hao and O'Shea 2011), where it was done on the background of msn4 deletion. Given the critical role of Msn4 found here, it is possible that this classification can change. The authors should address this point, either through data analysis/reanalysis or discussion of relevant caveats.*

We now include new Figure 5 to show single-cell gene regulation for other fast or slow promoters. To determine whether Msn2 and Msn4 exhibit distinct regulatory functions on promoters other than P*_DCS2_* and P*_SIP18_*, we examined fast kinetics promoter P*_DDR2_* and slow kinetics promoter P*_TKL2_*. As shown in Figure 5, P*_DDR2_* behaviors similar to P*_DCS2_* and P*_TKL2_* shows a similar response to that of P*_SIP18_*. In addition, to further examine the generality of distinct regulatory functions of Msn2 and Msn4, we analyzed gene expression under the mutated P*_SIP18_* promoter P*_SIP18_*-A4 (a slow promoter becomes a fast promoter; Hansen and O’Shea 2015) and the P*_DCS2_* promoter in cells lacking the SWI/SNF chromatin remodeling complex *(snf6△*) (a fast promoter becomes a slow promoter; Hansen and O’Shea 2013). As shown in Figure 5, P*_SIP18_*-A4 behaviors similar to P*_DCS2_*; P*_DCS2_* in *snf6△* shows a similar response to that of P*_SIP18_*. In summary, these results suggest that the distinct regulatory functions of Msn2 and Msn4 might depend more generally on the kinetics of promoter activation, but not on specific target promoters.

In the main text, we include a new section with two paragraphs describing the results:

“To determine whether Msn2 and Msn4 exhibit distinct regulatory functions on promoters other than P_DCS2_ and P_SIP18_, we measured the nuclear localization of Msn2 and Msn4 and reporter gene expression under fast kinetics promoter P_DDR2_ and slow kinetics promoter P_TKL2_. […] In summary, these results suggest that the distinct regulatory functions of Msn2 and Msn4 might depend more generally on the kinetics of promoter activation, but not on specific target promoters.”

Regarding a global view of gene regulation, we note that our single-cell analysis requires microfluidics-based live-cell time-lapse experiments, the throughput of which hinders the examination of gene regulation at a more global level. We include a paragraph discussing this limitation and also the influence of Msn4 on target gene classification in the main text:

“In this study, we have focused on a few representative promoters because our single-cell analysis requires live-cell time-lapse experiments, the throughput of which hinders the examination of gene regulation at a more global level. […] A genome-wide single-cell analysis will enable a more accurate classification of target genes and, more importantly, will lead to a comprehensive understanding about dynamic regulation of global transcriptional responses to environmental stimuli.”

*3) Static versus dynamic profiles of gene expression: The authors only report the final level of the target genes (at 30 or 60 minutes) but not the target genes dynamic profiles (as demonstrated in Figure 1, right, blue trace). With these data, the msn4 mutant has a non-responsive Sip18 after 30 minutes, yet responsive at 60 minutes. Previous data from the π (Hao & O'Shea 2012, Supplementary Figure 12) suggests Sip18 does respond and climb steadily in this strain, albeit more weakly/slowly. Is it possible then that the msn4 mutant kinetics of Sip18 (and other "slow" targets) is simply slower/weaker, rather than qualitatively different ("AND" gate becoming an "OR" gate at prolonged inputs, as suggested by the authors)? The authors should therefore report target gene single cell trace, and use it to provide a more holistic interpretation of the results shown in Figure 1, and Figure 3.*

We now include new Figure 1—figure supplement 1 to show the dynamic profiles of reporter gene expression in wild-type and mutants in response to 30 or 60-min inputs (old Figure 1—figure supplement 1 now becomes Figure 1—figure supplement 2). Because the dynamics of reporter expression are dominated by the timescales of translation and maturation of the reporter protein, we observed a similar time-dependent trend of reporter expression in wild-type and mutants; the difference is primarily on the final level of reporter expression, which represents the accumulated transcriptional activity.

We would also like to clarify that, as noted in the legend of Figure 1, we monitored single-cell gene expression responses over a 3-hour period, which is sufficient for the reporter to reach the plateau under all the conditions tested. We used the final level of gene expression at the end of the 3-hour experiment (not the level at the 30 or 60-minute time point; 30 or 60 minutes are the input durations) to represent the reporter responses in Figure 1, Figure 3 and Figure 4. The same quantification has been commonly used in previous papers (Figure 4 in Hao and O’Shea 2012, Hansen and O’Shea, 2013, 2015, 2016).

*4) Lack of quantitative data for some claims: Given the quantitative nature of this paper, some claims that could be made quantitatively are reported qualitatively. Results: "suggesting a competing role of Msn2 for binding to the promoter". The authors should make a plot to show the difference quantitatively.*

We include new Figure 3—figure supplement 3 to show the competing (or inhibitory) role of Msn2 against Msn4 quantitatively. In the figure, we quantified and plotted the relationships between gene expression and the ratio of Msn2 versus Msn4. We showed that gene expression decreases dramatically when the ratio of Msn2 versus Msn4 increases.

In the main text, we added “To quantitatively demonstrate this competing role of Msn2 against Msn4, we plotted the relationship between gene expression and the ratio of Msn2 versus Msn4 in single cells. As shown in Figure 3—figure supplement 3, gene expression decreases dramatically when the ratio of Msn2 versus Msn4 increases.”

*Similarly Results: "probability and the level of gene induction increase…". There is no data pointing to the "probability" (or at least it is close to impossible to figure out the probability from the distribution data).*

We include new Figure 3—figure supplement 2 to show the probability of gene induction (old Figure 3—figure supplement 2 now becomes Figure 3—figure supplement 4).

*Results: "much more heterogeneous". The authors should put numbers to the claim and probe whether this due to the relative small copy number of Msn4 compared to Msn2.*

To quantify the heterogeneity, we calculated the coefficient of variation (CV; standard deviation divided by mean) for the peak point of time traces and include these numbers above the plots in Figure 2. In addition, in the new Figure 2—figure supplement 2, we plotted single-cell time traces and averaged time traces with single-cell standard deviation after normalization of YFP and RFP. We also calculated the standard deviations of single cells for the peak time points (without being scaled by the means) and showed that although Msn4 has a higher CV, it has a lower standard deviation than Msn2. This suggests that the high degree of cell-cell variability can be largely accounted by the low mean level of Msn4.

In the main text we add: “and reported the coefficient of variation (CV: standard deviation scaled by the mean) for the peak point of time traces”.

We also add: “In addition, although Msn4 has a higher coefficient of variation, the standard deviation of Msn4 nuclear localization in single cells (without being scaled by the mean) is lower than that of Msn2 (Figure 2—figure supplement 2). These results suggest that the high degree of cell-cell variability of Msn4 might be largely due to its relatively low nuclear levels compared to that of Msn2.”

We removed “much”.

*Lastly, Discussion: "2-3min". If one normalizes the data (min to max) in Figure 2—figure supplement 1, Msn2 and Msn4 traces may simply overlay on top of each other (i.e., no time difference).*

We include new Figure 2—figure supplement 1, in which we normalize and overlay the time traces in Figure 2—figure supplement 1 (inhibitor input) as suggested and zoom in at the first 10 min of the response to show the time difference.

*5) Functional analysis of promoters: Based on the authors' analysis, it seems like slow and fast promoters are regulated differently and can have distinct physiological functions. To link these mechanistic findings to potential functional and physiological implications, the authors should provide analyses supporting such roles, for example, showing correlation between induction dynamics and regulation by Msn2/4 to functional classification. If no such relationship exists, authors should also indicate so. Related to this point, promoters with fast and slow activation kinetics have their own advantages during adaptation upon stress stimulation. The authors should include an introduction or conclusion discussing in some detail "how the two mechanisms mediated by the two kinds of promoters are important for adaptation…".*

We thank the reviewers for this suggestion. This is actually an analysis that we should have done years ago. Now we have performed the analysis and show the results in new Figure 6—figure supplement 1. We find a close correlation between gene functions and promoter kinetics for the target gene groups with fast or slow kinetics promoters (the gene groups are from Hao and O’Shea, 2012). We add a discussion of these results and the physiological implications (including the advantages of fast and slow kinetics) in the main text:

“Target genes with fast and slow promoter activation kinetics are regulated differently and hence might have distinct physiological functions. […] This temporal separation of target genes with different functions could avoid initiating resource-intensive cell protection processes in response to minor environmental fluctuations and thereby optimize resource allocation under rapidly changing environments.”

*6) Connection to osmotic stress: The reduced resistance of primed msn4 mutant cells to osmotic stress is intriguing, but the connection to the main story is not entirely clear. Should the authors opt to include this data, they should also include msn2 cells also respond to this stress under the same experimental conditions/protocols in order to probe whether this effect is Msn4 specific.*

We performed the priming experiments in msn2 cells and showed that, similar to msn4 cells, the effect of priming is also diminished in msn2 cells. This is an expected result in accordance with the gene expression data in Figure 1, in which both Msn2 and Msn4 are required for induction of slow kinetics genes. We add the msn2 data in the original figure (orange curves and boxes). These results demonstrate that the homologous TFs Msn2 and Msn4 are not redundant physiologically; instead they are both important for adaptive stress resistance.

Although these priming results clearly show that Msn2 and Msn4 are not physiologically redundant (one step further than gene expression), we do agree with the reviewers that these experiments are not very smoothly connected to the main story (which focuses primarily on single-cell gene regulation) in terms of experimental protocols and data presentation. Therefore, we deleted the paragraphs describing the results in the revised main text.